# DEEP REINFORCED ACTIVE LEARNING FOR MULTI-CLASS IMAGE CLASSIFICATION

## ABSTRACT

High accuracy medical image classification can be limited by the costs of acquiring more data as well as the time and expertise needed to label existing images. In this paper, we apply active learning to medical image classification, a method which aims to maximise model performance on a minimal subset from a larger pool of data. We present a new active learning framework, based on deep reinforcement learning, to learn an active learning query strategy to label images based on predictions from a convolutional neural network. Our framework modifies the deep-Q network formulation, allowing us to pick data based additionally on geometric arguments in the latent space of the classifier, allowing for high accuracy multi-class classification in a batch-based active learning setting, enabling the agent to label datapoints that are both diverse and about which it is most uncertain. We apply our framework to two medical imaging datasets and compare with standard query strategies as well as the most recent reinforcement learning based active learning approach for image classification.

## 1 INTRODUCTION

Modern methods in machine learning (ML), including deep learning (DL) frameworks, require large amounts of labelled data to train sufficiently well to obtain high performance. Depending on the training task, these data can be very expensive to obtain or annotate, to the extent that traditional approaches become prohibitively costly. Active learning (AL) aims to alleviate this problem by adaptively selecting training samples with the highest value to construct a minimal training dataset with the most information for the ML model.

In order to select training samples with the most information, different strategies are used in different AL cycles which can be either constructed based on knowledge of the specific problem one is aiming to learn, or using theoretical criteria to approximate mathematical bounds on information contained in the data. Standard query strategies in AL include the uncertainty-based approach (Lewis and Gale, 1994; Lewis and Catlett, 1994; Shannon, 1948; Scheffer et al., 2001; Esuli and Sebastiani, 2009; Seung et al., 1992; Dagan and Engelson, 1995), which aim to quantify the model uncertainty about the samples to be selected using different hand-crafted heuristics. Other approaches aim to estimate the expected model change (Roy and Mccallum, 2001; Freytag et al., 2014), or employ diversity-based approaches to promote diversity in sampling (Bilgic and Getoor, 2009; Gal et al., 2017; Nguyen and Smeulders, 2004).

Some approaches combine different techniques in hybrid-based query strategies, to take into account the uncertainty and diversity of query samples (Ash et al., 2019; Zhdanov, 2019; Shui et al., 2019; Beluch et al., 2018). Other methods leverage the exploration-exploitation trade-off and reformulate the AL framework as a bandit problem (Hsu and Lin, 2015; Chu and Lin, 2016) or a reinforcement learning problem (Ebert et al., 2012; Long and Hua, 2015; Konyushkova et al., 2017a), which are however still limited by their reliance on hand-crafted strategies, as opposed to learning a new one.

The move towards combining deep learning methods with active learning, to combine the learning capability of the former in the context of high-dimensional data, with the data efficiency of the latter, have led to further methods development. However, combining the two is non-trivial; traditional active learning query strategies label samples one-by-one, and so batch-model deep active learning aims to use batch-based sample querying (Gal and Ghahramani, 2015; Gal et al., 2017; Kirsch et al., 2019; Cardoso et al., 2017) to ensure efficiency in sampling the data.

Modern diversity-based approaches in deep active learning include the coreset approach (Sener and Savarese, 2018; Killamsetty et al., 2020; Wei et al., 2015; Shen et al., 2017; Mirzasoleiman et al., 2019), which aim to minimise the Euclidean distance between the sampled and unsampled data points in the latent space of the trained model. Whilst the coreset approach has been shown to work well for image classification tasks (Sener and Savarese, 2018), the performance deteriorates as the number of classes grows. Furthermore, as the dimensionality of the data grows, the distance measure between data points becomes indistinct due to the curse of dimensionality. Semi-supervised approaches (Sinha et al., 2019; Kim et al., 2020; Zhang et al., 2020) aim to alleviate this issue by using an adversarial network as a sampling strategy to pick data with the largest amount of information in the latent space.

Manually designing the DL models in addition to AL query strategies requires both expert knowledge about the task at hand, as well as a lot of compute resources to train the DL model. Furthermore, as the labelling heuristic is generally specific to the dataset of interest, there is little likelihood that the learnt acquisition function is transferrable to other datasets, whereas one based on a meta-learning approach may be more easily applied to other data domains.

In this paper we combine active learning, deep learning and reinforcement learning into an end-to-end framework which can automate the design of the acquisition function for active learning with high-dimensional, multi-class data, in a pool-based active learning setting. We additionally introduce a batched-labelling approach, enabling us to label multiple datapoints at each step, allowing for much more efficient training.

By employing coreset-inspired methods, we encourage the reinforcement learning agent to label samples which maximise uncertainty and are also diverse. We apply our model to medical image classification datasets, covering binary and multi-label classification problems as well as differing imaging modalities. We add a range of noise to the images, in order to simulate a real-world annotation setting, and show that our framework is robust to high levels of noise. We compare the classification accuracies to a wide range of sampling methods, including (Konyushkova et al., 2018), the most similar framework to ours, and we show we outperform all other strategies.

## 2 RELATED WORK

Recent papers on combining active learning with reinforcement learning aim to instead learn a policy for labelling data from the unlabelled pool in order to maximise model performance. In (Bachman et al., 2017; Liu et al., 2018), they use information gathered from an expert oracle to learn the policy, whilst (Pang et al., 2018; Padmakumar et al., 2018) use policy gradient methods to learn the acquisition function. Current papers which aim to combine deep reinforcement learning with active learning are not able to label more than one datapoint per step (Konyushkova et al., 2018; Woodward and Finn, 2017; Pang et al., 2018; Padmakumar et al., 2018; Bachman et al., 2017; Liu et al., 2018; Woodward and Finn, 2017), with the exception of (Casanova et al., 2020) which labels batches of pixels for semantic segmentation tasks.

Many existing works on reinforced active learning focus on the simpler task of a stream-based active learning approach (Fang et al., 2017; Woodward and Finn, 2017) or are limited to binary tasks (Konyushkova et al., 2018; Pang et al., 2018; Liu et al., 2019), which limit their application to more general, and difficult classification tasks. In (Haussmann et al., 2019), they learn an acquisition function using a Bayesian neural network which is layered onto a bootstrapped existing heuristic.

In this work we focus on medical image classification tasks, which has attracted attention in the field of deep active learning, due to the cost of acquiring medical image data as well as the relatively small size of the datasets. Image segmentation tasks in active learning have traditionally used hand-crafted uncertainty-based acquisition functions (Wen et al., 2018; Smailagic et al., 2018; Konyushkova et al., 2016; Gal and Ghahramani, 2015; Gal et al., 2017; Yang et al., 2017; Ozdemir et al., 2018). Generative adversarial network (GAN) based methods, used widely for image synthesis, have been used in order to add informative labelled data to limited training sets, which is directly applicable for active learning scenarios (Zhao et al., 2019; Mahapatra et al., 2018; Last et al., 2020). There recently has been some research applying meta-learning to medical image tasks in an active learning setting; in MedSelect (Smit et al., 2021) they use reinforcement learning to label medical images, and in (Konyushkova et al., 2017b) they use a regression model to learn a query strategy using greedy selection.

Our work is closest to (Konyushkova et al., 2018), in that we use a double deep Q-network (van Hasselt et al., 2015) (DDQN), a generalisation of the deep Q-network (Mnih et al., 2015) to learn the acquisition function and to avoid bias from overestimation of the action values and stabilise training, in a pool-based setting. Our work extends that of (Konyushkova et al., 2018) in several ways. Firstly, we show that we are not limited to binary classification tasks and can accurately predict in a multi-class, high dimensional setting, by integrating a deep learning approach into the framework. Secondly, we use batch-based active learning to label multiple datapoints at a time, thus greatly improving on the efficiency of the framework. We also aim to perform the data labelling in a much more realistic environment, and so we add realistic levels of noise to our data, specific to the differing image modalities, to simulate a real-world setting where the images are distorted.

## 3 METHODOLOGY

As stated above, our approach aims to reformulate the active learning framework as a reinforcement learning problem, by casting pool-based active learning as a Markov decision process (MDP), inspired by (Konyushkova et al., 2018) and (Casanova et al., 2020) and then use a reinforcement learning agent to find an optimal labelling strategy. By using not only a meta-learning approach, but a full reinforcement learning approach, the selection strategies are learnt in a data-driven way, thus removing the bias incurred from using hand-crafted acquisition functions.

### 3.1 ACTIVE LEARNING AS A REINFORCEMENT LEARNING TASK

In order to use reinforcement learning to learn a query strategy, we need to cast active learning as a MDP. An MDP is defined by the tuple $\{(s_t, a_t, r_{t+1}, s_{t+1})\}$. For each state $s_t \in \mathcal{S}$, the agent can perform actions $a_t \in \mathcal{A}$ to choose which sample(s) to annotate from an unlabelled dataset $\mathcal{U}_t \in \mathcal{D}$. The data are then added to the labelled dataset $\mathcal{L}_t \in \mathcal{D}$. Let $f_t$ be a classifier (in our case, a convolutional neural network) trained on the labelled data $\mathcal{L}_t$. The action $a_t$ is then a function of $f_t, \mathcal{L}_t$ and $\mathcal{U}_t$. Depending on the sample(s) chosen to be labelled by the agent, it then receives a reward $r_{t+1}$.

We can then formulate active learning as an episodic MDP. We initialise the active learning loop with a small labelled set $\mathcal{L}_0 \in \mathcal{D}$ alongside the unlabelled dataset $\mathcal{U}_0 = \mathcal{D} \backslash \mathcal{L}_0$ where $|\mathcal{U}_0| \gg |\mathcal{L}_0|$. At each iteration we then

1. Train the classifier $f_t$ on the initial labelled data $\mathcal{L}_0$.
2. Compute the state $s_t$ as a function of the prediction of the CNN $f_t : \hat{y}_t(x_i) \mapsto \hat{y}_i$, as well as $\mathcal{L}_t$ and $\mathcal{U}_t$.
3. The active learning agent selects $N$ actions $\{a_t^n\}_{n=1}^N \in \mathcal{A}$ by following a policy $\pi : s_t \mapsto a_t$ that defines data $x_t \in \mathcal{U}_t$ to be labelled. Each action $a_t^n$ corresponds to one image to be labelled.
4. An oracle labels the data with $y_t$ and updates the sets $\mathcal{L}_{t+1} = \mathcal{L}_t \cup \{(x_n, y_n)\}_{n=1}^N, \mathcal{U}_{t+1} = \mathcal{U}_t \backslash \{x_n\}_{n=1}^N$.
5. Train $f_t$ one iteration on the new labelled dataset.
6. The agent receives a reward $r_{t+1}$ based on the performance of $f_{t+1}$ on a test set.

The episodes terminate when they reach a terminal state, which is when the labelling budget is met. Once the episode is terminated, we reinitialise the weights of the classifier as well as the labelled and unlabelled datasets. The aim of the agent is to learn a policy $\pi$ to maximise the expected discounted reward until some terminal state is reached

$$Q^\pi(s, a) = \mathbb{E}\left[\sum_{i=0} \gamma r_i\right], \tag{1}$$

where we set the discount factor $\gamma = 0.99$. The optimal value is then $Q^{\pi*}(s, a) = \max_\pi Q^\pi(s, a)$. We compute the reward at each time-step as the difference between the accuracy at the current and previous time-steps of the prediction of the classifier on a hold-out set $\mathcal{D}_R$.

$$r_{t+1} = \text{ACC}(\hat{y}_{t+1}(\tilde{x}_i), \tilde{y}_i) - \text{ACC}(\hat{y}_t(\tilde{x}_i), \tilde{y}_i), \ \tilde{x}, \tilde{y} \in \mathcal{D}_R. \tag{2}$$

### 3.1.1 State representation

Following (Konyushkova et al., 2018) we represent the state space $\mathcal{S}$ using a set-aside dataset $\mathcal{D}_s$. We use the sorted list of predictions of $f_t$ to represent $s_t$. As we use a CNN for our classifier, unlike (Konyushkova et al., 2018) where they use logistic regression or a support vector classifier, we can represent the state for multi-class data and are not limited to binary tasks.

### 3.1.2 Action representation

In the active learning setting, taking an action determines which datapoint(s) to label from the unlabelled pool. We generalise the standard DDQN architecture, by representing each action $a_t^n$ as a concatenation of 3 features: the score of the classifier $f_t$ on $x_n$ and measures of distance between $x_n$ and the labelled and unlabelled datasets in the latent space of the classifier. This enables us to label datapoints explicitly requiring diversity of samples, and not just based on how uncertain the classifier is about the potential new datapoint. We compute distance measures in the feature space of the classifier inspired by the coreset approach of (Sener and Savarese, 2018), which, while coresets have been shown to be ineffective for high-dimensional and multi-class data, we find that by including latent space distance measures, as opposed to distance measures directly in the space of the data, that training is substantially faster for little reduction in accuracy.

## 3.2 Learning a policy for data annotation

We employ a DDQN (van Hasselt et al., 2015), parameterised by a multi-layered perceptron, $\phi$, to find $\pi*$. We train the DDQN with a labeled training set $\mathcal{D}_T$ and compute rewards using a hold-out set $\mathcal{D}_R$.

Similarly to (Casanova et al., 2020), we aim to annotate the unlabelled data at each step in a data-efficient manner, by labelling batches of data at a time. As we are focused on image classification, as opposed to semantic segmentation, we do not consider regions of images to be labelled, rather we wish to label entire images at each step. We can therefore label $N$ images by calculating the top-$N$ $Q$-values at each step

$$\{a_t^n\}_{n=1}^N = \underset{a_t^n \in \mathcal{A}, |a_t^n|=N}{\arg\max} \sum_{Q^\pi \in a_t^n} Q(s_t, a_t^n; \phi). \tag{3}$$

To stabilise training and alleviate the overestimation bias of the DQN framework, we use a target network with weights $\phi'$ using an adapted DDQN architecture, to enable us to work with actions represented by vectors, and the network is trained using the temporal difference (TD) error (Sutton, 1988)

$$y_t = r_{t+1} + \gamma Q \left( s_{t+1}, \underset{a_t^n \in \mathcal{A}, |a_t^n|=N}{\arg\max} \sum_{Q^\pi \in a_t^n} Q(s_t, a_t^n; \phi'); \phi \right). \tag{4}$$

## 4 Experiments

In this section we wish to evaluate our method against different medical image datasets, which span differing imaging modalities. We will introduce the datasets we use, as well as the experimental set-up and baselines against which we compare.

## 4.1 Baselines

We will compare our method with several baselines. Firstly, as our work can be considered a generalisation and extension of LAL-RL (Konyushkova et al., 2018), we compare our results against theirs. We then compare our method with 'standard' active learning query strategies, implemented using the `modAL` package (Danka and Horvath). Namely, the baselines we compare against are

- **LAL-RL** (Konyushkova et al., 2018), using logistic regression as the base classifier, as done in the original work.
- **Random sampling**, where the datapoint to be labelled is picked at random.

- **Margin sampling** (Scheffer et al., 2001), which selects the instances where the difference between the first most likely and second most likely classes are the smallest.

- **Entropy sampling** (Shannon, 1948) , which selects the instances where the class probabilities have the largest entropy.

- **Uncertainty sampling** (Lewis and Gale, 1994) , which selects the least sure instances for labelling.

- **Average confidence** (Esuli and Sebastiani, 2009), a query strategy designed for multi-label classification.

For the baseline query strategies except for LAL-RL, we use the same CNN classifier as the active learner as in our implementation, which is a pre-trained resnet50 (He et al., 2015) model with a linear layer added to ensure the correct number of labels are classified for a given dataset.

## 4.2 IMPLEMENTATION DETAILS

As we use a pre-trained resnet50 CNN as the classifier for our approach and those all benchmark methods except LAL-RL, we apply transforms to the images in the differing datasets consistent with the image size and normalisations required (see paper for full details, (He et al., 2015)). In order to ensure consistency across the benchmarks we apply these transformations to the data also when comparing against LAL-RL.

To compute $Q^\pi(s, a)$ we use a multi-layered perceptron with 3 hidden layers and 128 neurons in each layer with ReLU activation function. We train the DDQN with a fixed learning rate of 0.0001, requiring the DDQN to train until the average reward obtained has stabilised using early stopping. As stated above, the classifier, $f$, is the standard resnet50 CNN classifier with an additional linear layer to enable multi-label classification, which we train for 200 epochs with a fixed learning rate of 0.0001.

Due to the similarities between our approach and those of (Konyushkova et al., 2018), all the overlapping reinforcement learning hyperparameters are kept consistent when running the experiments. In practice, this means that the classifier and action representations are the main parameters which are changed between the two methods. Due to the very large compute resources required by LAL-RL, as we will discuss in detail below, we only initialise the DDQN for both methods with 16 warm start random episodes. For each method, we show results on the accuracy of the classifier on the test set for each dataset, after having been trained on a set number of datapoints. We show results for 5 different random initialisations, showing the mean and 68% confidence level intervals for all methods.

## 4.3 DATASETS

We evaluate our method on two open source medical imaging datasets under CC BY 4.0 licence[1], with an aim to cover very different imaging modalities and types of data present in medical image classification tasks, and to look at the binary and multi-label classification scenarios.[2]

**Pneumonia**   The first dataset consists of chest X-rays of paediatric patients (Kermany et al., 2018a;b) with the classification task being to determine whether the patient presents with pneumonia or not.

**Colorectal cancer**   This 8-class classification problem consists of histopathology tiles from patients with colorectal adenocarcinoma (Kather et al., 2016a;b). The data consists of 8 differing types of tissue.

## 4.4 RESULTS

We show in Fig. 1 the accuracy of the differing query strategies as a function of labelled datapoints for the pneumonia (left) and colorectal cancer histopathology (right) datasets. Error bands are the

---

[1]https://creativecommons.org/licenses/by/4.0/legalcode

[2]The human biological samples were sourced ethically and their research use was in accord with the terms of the informed consents under an IRB/EC approved protocol.

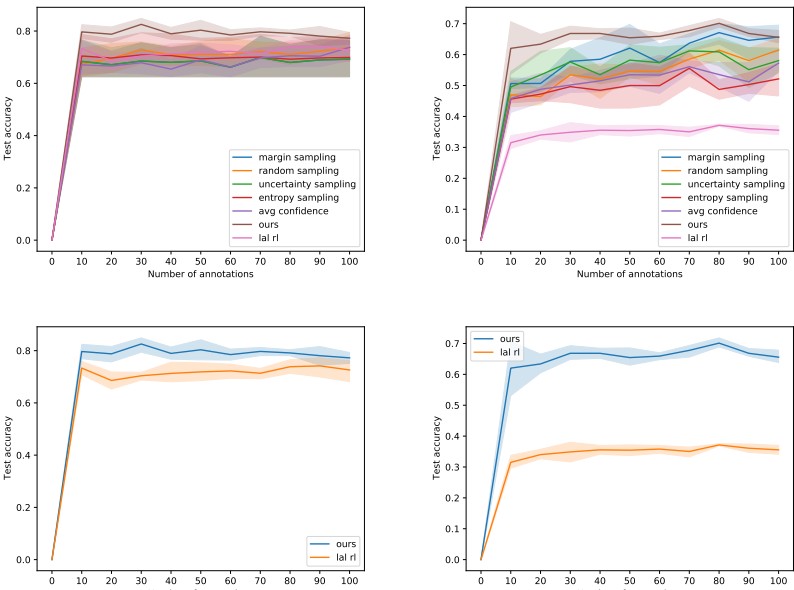

Figure 1: Top: Accuracy on the test set as a function of labelled datapoints for the pneumonia (left) and colorectal cancer histopathology (right) dataset for the differing labelling strategies. Error bands are the 68% confidence level intervals over random initialisations of 5 seeds. Bottom: Same as before but with just the results of the two reinforcement learning approaches.

68% confidence level intervals over random initialisations of 5 seeds. As we wish to emphasise the generalisation of our method over existing reinforcement learning based active learning query strategies, we plot the same results in the bottom panel for our method and LAL-RL.

We observe that, regardless of query strategy, our method on average outperforms all others, importantly reaching near-optical accuracy on the test set for the colorectal cancer histopathology dataset with very few labelled datapoints. This suggests that, because our method is substantially more general than the other strategies, it is able to better deal with the multi-class labelling task for the histopathology slides. Importantly, our method outperforms LAL-RL on the binary classification pneumonia dataset, for which LAL-RL is designed to work with, whilst ours is agnostic to the number of classes present in the data. The standard deviation of results both in our approach and LAL-RL are also much smaller than those of the traditional querying methods, suggesting that a reinforcement learning based approach is in fact more stable with respect to random initialisations of the dataset and hyperparameters.

A fundamental difference between our approach and that of LAL-RL is how we compute the generalised action representation; i.e., how one determines the distances between the labelled and unlabelled datasets. Our approach is inspired by coreset methods (Sener and Savarese, 2018), where we minimise the Euclidean distance between the sampled and unsampled data points in the latent space of the trained model. LAL-RL, however, compute the pairwise distances in the space of the data, which, in the case of very high dimensional data as we have here, can become computationally intractable. Indeed, our experiments show that training the colorectal cancer histopathology dataset for 5 episodes, allowing 20 datapoints to be annotated with the two reinforcement learning methods with identical overlapping hyperparameters is 267 times slower using LAL-RL than our method.

### 4.4.1 COMMENT ON BENCHMARK RESULTS

As we mentioned above, this extremely slow performance requires us to use very lightweight models for the two reinforcement learning based approaches in order to be able to benchmark between them, preventing the true predictive power of either of them from being accurately represented. We therefore emphasise that the results we show in this section are only indicative of the general improvement of

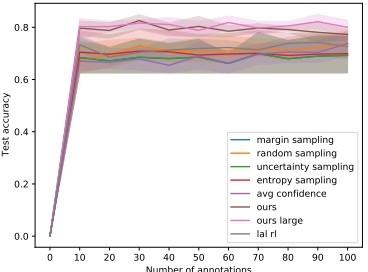 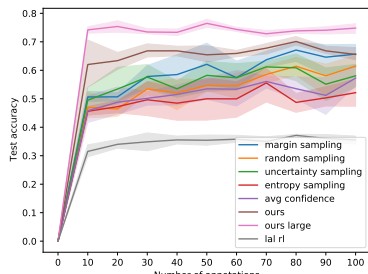

Figure 2: Same as in Fig. 1 but additionally showing results for the two datasets with a slightly larger model for our method.

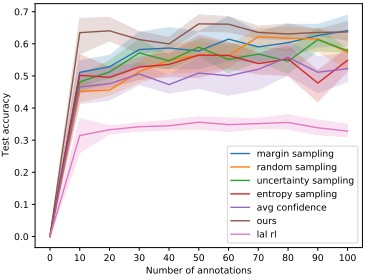 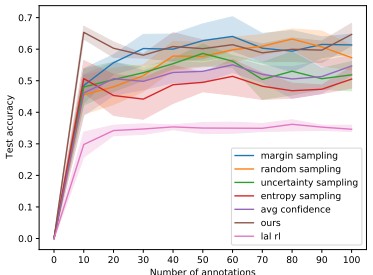

Figure 3: Test accuracy on the colorectal cancer histopathology histopathology dataset where training noise has been added to 10% (left) of the dataset and 100% (right).

our approach over other query strategies and by no means the best predictive power possible for our method on the two datasets we show in this work.

Indeed, by simply increasing the number of warm start episodes from 16 to 128 for our method, so that the replay buffer is filled with a better selection of episodes from which to sample as well as adding one additional datapoint per step using Eq. equation 3, and keeping all other parameters the same, we find up to 11.5% improvement in test accuracy for the pneumonia dataset and 23.3% for the colorectal cancer histopathology dataset, as we show in Fig. 2 where we additionally plot the results for this slightly larger model. We see that in the multi-class case, the small increase in size of our DDQN drastically improves performance on the dataset, as well as greatly reducing the spread of results and is thus much more stable. We can therefore conclude that our method can easily outperform other active learning query strategies in terms of: predictive power, stability, and efficiency, as (near-) optimal performance is found by labelling very few data points.

### 4.4.2 ROBUSTNESS TO NOISE

In this section we modify the training data to account for noise in the data. We randomly augment the training images by randomly zooming the images, rotating them by up to $\pm\pi/12$ radians, and adding random Gaussian multiplicative noise using the MONAI package (MONA, 2022), a medical imaging deep learning package which provides image transforms of relevance to differing image modalities. We ran experiments for the aforementioned noise applied to 10% of the training set as well as to 50% and 100%, to cover a range of eventualities.

In Fig. 3 we again show the test accuracy on the colorectal cancer histopathology dataset for the differing query strategies where training noise has been added to 10% (left) of the dataset and 100% (right). It is clear that all query strategies are, to an extent, robust to noise in the data, which suggests that there is no overfitting in any of the cases. We quantify this in Table 1 where we show the accuracy on the test set for differing query strategies after labelling 10 datapoints with differing levels of noise added to the training data and in all cases the mean accuracy is consistent within the error bands.

| Strategy | No noise | 10% | 100% |
|---|---|---|---|
| Ours | $0.6202 \pm 0.0981$ | $0.6348 \pm 0.0638$ | $0.6534 \pm 0.0226$ |
| LAL-RL | $0.3151 \pm 0.0238$ | $0.3143 \pm 0.0633$ | $0.2978 \pm 0.0463$ |
| Uncertainty sampling | $0.4952 \pm 0.0493$ | $0.4813 \pm 0.0285$ | $0.4802 \pm 0.0670$ |
| Random sampling | $0.4698 \pm 0.0540$ | $0.4516 \pm 0.0385$ | $0.4552 \pm 0.0697$ |
| Margin sampling | $0.5063 \pm 0.0204$ | $0.5111 \pm 0.0435$ | $0.4849 \pm 0.0628$ |
| Entropy sampling | $0.4560 \pm 0.0168$ | $0.5020 \pm 0.0673$ | $0.5067 \pm 0.0589$ |
| Average confidence | $0.4579 \pm 0.0539$ | $0.4655 \pm 0.0630$ | $0.4611 \pm 0.0731$ |

Table 1: Accuracy on the test set for the colorectal cancer histopathology dataset for differing query strategies after labelling 10 datapoints with differing levels of noise added to the training data.

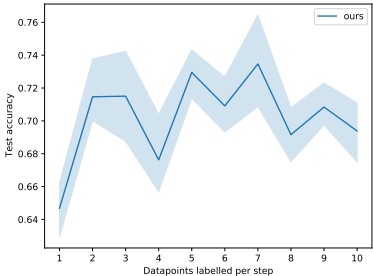

Figure 4: Accuracy on the test set as a function of the number of datapoints labelled at each step, corresponding to the top-$N$ $Q$-values as determined by the reinforcement learning algorithm.

### 4.4.3 PICKING N-DATAPOINTS AT A TIME

As we discuss above, one of the main advantages of our method is the ability to label multiple datapoints at each stage of the active learning cycle. We do this by labelling the top-$N$ $Q$-values at each step, following Eq. equation 3. It is natural to consider several consequences of labelling data in this way. Firstly, by labelling lower-ranked datapoints at each step, one may ask whether the quality of prediction of the model deteriorates as we increase $N$. Secondly, does the model require any additional time to perform these additional labellings, as the size of the training set now is allowed to increase.

To answer these questions, we ran our DDQN model on the colorectal cancer histopathology dataset, training for a fixed number of episodes and allowing 50 total label annotations each episode. At each step, we allow the agent to label from $\{1..10\}$ datapoints, and as before, compute the accuracy on the test set. We chose the colorectal cancer histopathology dataset as here the contrast between the other reinforcement learning informed query strategy, LAL-RL, is most pronounced. We show in Fig. 4 the mean and 68% confidence level intervals, averaged over 5 seeds, of this procedure. It is clear that labelling more than 1 datapoint per step can greatly improve the prediction accuracy of the model, suggesting our prior results in Sec. 4.4 are *conservative* with respect to the potential prediction power of our model. Beyond allowing $N > 1$, however, within the error bands as reported, it is not obvious that labelling more data at each step *for this dataset* helps (or indeed, hinders), prediction accuracy.

The training time for the differing values of $N$ was also computed, and we show the test accuracies for the differing values of $N$ as shown in Fig. 4 alongside the average time for the training in seconds. We can see that the training time does not begin to deteriorate until $N > 8$, which as we can see, does not even come at the gain of improving test accuracy for this dataset. It is therefore reasonable to consider $1 < N < 9$ to be a logical choice of datapoints to label in this experiment.

## 5 DISCUSSION

In this paper we have introduced a novel method for active learning query strategies, inspired by previous work which reformulates the active learning cycle as a Markov decision process. Similarly to previous work (Konyushkova et al., 2018), we use a deep reinforcement learning approach to cast the task of labelling data as actions for an agent to pick in an environment. Unlike previous reinforcement learning based active learning query strategies, however, we aim to present a much

| $N$ | Test acc | Time (seconds) |
|---|---|---|
| 1 | $0.6468 \pm 0.0197$ | 1511 |
| 2 | $0.7147 \pm 0.0220$ | 1633 |
| 3 | $0.7151 \pm 0.0307$ | 2505 |
| 4 | $0.6762 \pm 0.0272$ | 3313 |
| 5 | $0.7295 \pm 0.0168$ | 2123 |
| 6 | $0.7091 \pm 0.0185$ | 1222 |
| 7 | $0.7347 \pm 0.0321$ | 1255 |
| 8 | $0.6916 \pm 0.0203$ | 1290 |
| 9 | $0.7084 \pm 0.0148$ | 3374 |
| 10 | $0.6939 \pm 0.0206$ | 6504 |

Table 2: Test accuracy and training time for our model labelling $N$ datapoints at each step.

more general approach to active learning, by enabling multi-label prediction as opposed to just binary classification. In addition, by enabling the agent to pick an arbitrary number of datapoints at each step in the MDP, our approach can easily be applied to segmentation tasks where we would like to be able to label multiple pixels at each step, such as in (Casanova et al., 2020).

We have applied our method to the task of medical image classification, in the binary and multi-label class case, and across differing image modalities. We have shown that, even with extremely conservative estimates, our approach outperforms both standard query strategies, as well as the current state of the art reinforcement learning based approach, LAL-RL. Importantly, our method has been shown to be much more computationally efficient, as optimal accuracy on the test set can be reached with very few labelled data points, and by sampling the data in the latent space of our classifier, requires much less compute resources than LAL-RL. This is a very relevant result in the space of medical imaging, as, depending on the imaging data, one may be limited by the number of samples one can obtain but each datapoint is very high-dimensional. In the healthcare domain, high-$d$, low-$n$ datasets are abundant, whilst traditional active learning-based approaches generally assume one can have access to a large pool of unlabelled data.

Both LAL-RL and our method aim to answer the same question; namely, can we create an active learning framework which contains minimal bias in how data is labelled at each stage? We take this one step further and consider; can we create an active learning framework which can do this task but apply it to the deep learning paradigm of active learning? In this paradigm, we do not wish to limit ourselves to binary classification tasks, we wish to be able to add large batches of data at each step as opposed to just one datapoint, and we also need to be able to deal with high-dimensional data. We solve all these problems in our approach: in the first instance, we can clearly outperform LAL-RL on an eight-label classification task. In the second instance, by ranking the top-$N$ $Q$-values at each step, we show we obtain good performance (at improved accuracy and minimal loss of speed), and finally, by using a coreset-inspired approach for determining the action representation, we are much more able to deal with very high dimensional data.

## 5.1 Outlook

There are several avenues for exploration to extend this work. We mention above that image segmentation would be an interesting application of our approach; indeed, although here we just consider image classification, our framework is task-agnostic and can in theory be applied to arbitrary active learning cycles. As we discuss above, although we have empirically seen that our coreset-inspired approach to ranking $Q$-values works well, it is known that this method breaks down in the multi-label regime, and so determining a heuristic which is more efficient for multi-label classification would of be importance to improve performance of our method.

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
