# OpenReview forum: "Deep reinforced active learning for multi-class image classification"
_ICLR.cc/2023/Conference — Submitted to ICLR 2023_

### Official Review · Reviewer_CZBN · 2022-10-19

**Confidence:** 5
**Correctness:** 2
**Technical Novelty And Significance:** 2
**Empirical Novelty And Significance:** 1
**Recommendation:** 3

**Clarity, Quality, Novelty And Reproducibility:**

- There are typos, language sometimes sub-optimal, please proof read.
   - “a[n] MDP”
   - “Eq. equation 3”
   - “near-optical”
   - “we are much more able to deal with..”
   - Training time in Table 2 is referenced in text as Fig. 4.
   - $\pi{\star}$ $\rightarrow$ $\pi^\star$
   - By a multilayer perceptron, $\phi$, $\rightarrow$ with parameters $\phi$
   - Use \citep and \cite depending on how you use the reference in the paper
- What is the motivation to apply noise to the data instead of to the oracle? Have you considered noisy oracles?
   - Can you explain why noise on 100% of the training dataset still maintains the same accuracy on the test dataset?

Novelty:
- The claimed contributions are in the formulation of the action and in the top-n batch size operation. So please describe the difference to the policy that learns the selection of a batch via the top-n operation, that can be found in [Loeffler 2022; ref. see above]


**Strength And Weaknesses:**

**Strengths:**
- The paper is (at least partially) well-written and easy to
- The problem is interesting and relevant to the community

**Weaknesses**:
- Novelty is weak, the top-n operator was previously proposed for deep active learning [1] and also being trained within an RL-style framework
- The mathematical notation is inconsistent
- Several central claims are not supported by the experiments: (i) policy learning diversity sampling (I doubt this, as I cannot see this in the introduced methodology), (ii) that the learnt acquisition function can be transferred between datasets, (iii) that only little computational resources are required in relation to current approaches.
- The experimental study has several weaknesses (see more details below).


****Methodology****

- It is not clear how you minimize the Euclidean distance, as you say in Section 4.4 (last paragraph). Isnt it just a part of the action
- The paper introduces the MDP using a four-tuple $\{(s_t, a_t, r_{t+1}, s_{t+1})\}$ – usually that is not correct as an MDP is defined over a set and not over concrete time-step samples, i.e., { $\mathcal{S},\mathcal{A},\mathcal{R},[\mathcal{P}], \gamma$ }, where $\mathcal{S}$ is a set of states, $\mathcal{A}$ a set of possible actions, $\mathcal{R}$ a reward function, $\mathcal{P}$ a probability matrix that describes the system dynamics, and $\gamma$ the discount factor. Elements such as $s_t$ or $a_t$ are concrete elements from those sets at particular time-steps but they do not define the MDP.
- “5. Train f_t one iteration on the new labelled dataset” – what do you mean with one iteration? One epoch? If not: then I doubt that we see a full RL setting here. Does it make a difference if we label datapoint A first and then B versus labeling B first and then A?
- Equ. 1 is incorrect: $\gamma$ should be to the power of $i$. How is $i$ bounded? (I guess over the length of the episode, but there is a lot of notation being inconsistent here…)
- $Q^{\pi\star}$ $\rightarrow$ $Q^{\pi^\star}$
- “The optimal value is the $Q^{\pi^\star} (s,a)$ = $max_\pi Q^\pi (s,a)$ – please refer to standard RL literature to clear abuse of notation
- State representation: I have two questions on that: (1) Don’t you include any classifier-model-specific parameters into the state? If not, I doubt that the model explicitly learns diversity-based strategies…, (2) The state representation seems to be a bit misaligned and too minimal. I guess you formulate a POMDP here.
- Sec. 3.1.2 was unclear to me for quite some time and only became clear after iterating together with Section 3.2. The authors should rewrite this part to make it more accessible and more intuitive.
- Equation 4 is unclear. The argmax-operator as defined (although I am not sure if this is common-sense to do it like this) returns a set of actions but it is used as a scalar for computation. This is undefined. Or do you use the set over A as a proxy for actions? But then it is inconsistent with the description in 3.1.2

**Experimental results**

- Claims of superior performance should be better supported with suitable stronger baseline methods in experiments section (Coreset [Sener 2018], Ensemble [Beluch 2018], MC-Dropout [Gal 2017], IALE [Loeffler 2022], BADGE [Ash 2020 (please see the final reference paper instead of your cited preprint], BatchBALD [Kirsch2019, please see the final reference paper instead of your cited preprint], ...).
- No ablation study of the differences between Konyushkova et al. (2018) and the proposed method. That would be interesting to see.
- Why do you show only the 68% confidence interval for your results?
- Please define what N is in the main experiments (I guess it is 4)
- Figure 1ff: the accuracy at point 0 (annotions = 0) seem to be wrong. Is it the performance of the classifier after the warm-up? (then, I guess it has already seen 64 samples (16 episodes à 4 samples). Shouldn’t the graphs start at 64? At x=10 the classifier has seen 74 samples, right? (64 random + 10 selected by heuristic)
- To make your method more comparable, how does your method perform compared to the baselines on commonly used image classification datasets such as CIFAR10/100, SVHN, and MNIST variants, with commonly used classifier architectures (ResNet18 etc)?
- The presentation of results in Figs. 1-3 is unintuitive and there is little to no benefit to active learning visible. Why is that? I would also expect to run AL at least for 1K annotations to see how things evolve…
- How do the warm-start episodes exactly work? This seems important, can you perform a study on the effect and cost of it?
- The experiments on batch-sizes between 1 and 10 have several issues:
   - Is each datapoint in Figure 4 a separate run of the policy from 0 to 50 samples? If yes, why is the graph in Fig. 4 connecting the separate runs as if they are continuous? I’d like to see each run from 0 to 50 over each other in one figure.
   - What is the warmup?
   - Training time measurements seem noisy between 1222 and 3374, the N=10 datapoint seems like an outlier from this, how does it perform for N=20, 50, 100, 1000?
   -Why is 1 worse than the others?
- To me runtime or computation time seems to be a real issue. In fact the authors also mention that throughout the paper but there is no results or analysis on that.
- Section 4.2.2: to me this section is a bit misleading. I do not see why this aspect seems to be important here.
- What about model transfer, i.e, learning the policy on one dataset and the applying it to another?

**Some minor points to the experiments**
- How do you differentiate from MedSelect?
- The results in Fig. 1,2,3 do not present the results well, due to the point at 0 labeled samples that can be removed. Also, no sub-figs used there.
- Table 1: it would help to highlight the best-performing method

**Missing references and prior work & updated citations**
- [Loeffler 2022] Christoffer Loeffler and Christopher Mutschler: “IALE: Imitating Active Learner Ensembles”. In: Journal of Machine Learning Research 23 (2022) 1-29
- [Ash 2020] Jordan T. Ash, Chicheng Zhang, Akshay Krishnamurthy, John Langford, and Alekh Agarwal. Deep batch active learning by diverse, uncertain gradient lower bounds. In International Conference on Learning Representations (ICLR), 2020.
- [Kirsch 2019] Andreas Kirsch, Joost van Amersfoort, Yarin Gal. BatchBALD: Efficient and Diverse Batch Acquisition for Deep Bayesian Active Learning. NeurIPS, 2019.

**Summary Of The Paper:**

The authors propose deep-Q learning to learn an active learning strategy for medical image classification using deep learning models. The authors propose to add a geometric distance in the embedding space to the actions and then select the top-n Q values to compose the batch in bool-based batch-mode active learning. They claim that this (action/problem) formulation leads to uncertain and diverse batches.

**Summary Of The Review:**

This review considers the claims wrt. the conducted experiments and uncited related work. The experiments are not supporting the claims, and the there is no theoretical justification for them either. Furthermore, the novelty is doubtful given prior publications, and there are many other issues with the submission.

---

### Official Review · Reviewer_6vUj · 2022-10-23

**Confidence:** 4
**Correctness:** 2
**Technical Novelty And Significance:** 2
**Empirical Novelty And Significance:** 2
**Recommendation:** 3

**Clarity, Quality, Novelty And Reproducibility:**

Novelty: There seems to be significant overlap with this submission and Konyushkova et al. 2017b, 2018. I understand difference with these works is that the base classifiers used in Konyushkova et al. 2017b, 2018 were not deep nets. However, simply replacing the base classifier with another is not sufficient for an ICLR admission. The paper should also address why using a deep net with the remaining reinforcement framework was challenging. Why the additional steps that had to be taken, if any, to use deep net as base classifier make sense and sets the submission apart from Konyushkova et al. 2017b, 2018?

Experimental justification: As a reader I also need to understand why reinforcement learning for active learning with deep nets is a good approach in comparison to other active learning algorithms for deep nets. As such, comparison with existing DL active learning methods such as coreset (Sener and Savarase 18), VAAL (Sinha et al 19) and others must be included in the experiments section. Pointing out a few other works that could also be compared/explored for connection:
1. One bit active query with contrastive pairs, CVPR 21
2. Learning loss for active learning, CVPR 19
3. Active continual fine tuning for CNNS, MIA 21

Might also consider narrowing down the scope for cold or warm start and compare with prior arts accordingly.

Generality: Based on the submission, we have no evidence that the method would work for any task on natural images. It might be easier to convince the generality of this work if the performance was reported on natural images.

**Strength And Weaknesses:**

A principled framework, that is also learnable, for active learning would be very useful for active learning in my opinion.

**Summary Of The Paper:**

The paper proposes a reinforcement learning framework for active learning for deep learning where the examples to be annotated are selected by a policy. The policy is learned by a DQN to maximize (discounted) reward. The paper seems to be built upon works of Konyushkova et al. 2017b, 2018.

**Summary Of The Review:**

Although the proposed method his promising, I do not think the current manuscript is ready for publication at a major venue yet. The paper needs to clarify its novelty and conduct extensive experiments to show its merit on real data (preferably natural images).

---

### Official Review · Reviewer_tpMj · 2022-10-24

**Confidence:** 4
**Correctness:** 2
**Technical Novelty And Significance:** 2
**Empirical Novelty And Significance:** 3
**Recommendation:** 3

**Clarity, Quality, Novelty And Reproducibility:**

On clarity, I find the work reasonably clearly written. However, there are aspects of the results that are somewhat odd and would benefit from further insights.

On quality, I am not convinced by the experimentation done here. I would have liked other datasets to be included given the flat-lining mentioned above. I would also have liked different aspects of the approach to have been investigated in isolation, to clarify the degree to which they contribute to the results.

On novelty, as mentioned above I find it a bit weak.

On reproducibility, code to reproduce the experiments would help.


**Strength And Weaknesses:**

Strengths
---------
- The paper addresses the important and relevant topic of how to minimize the amount of labelled data needed for training deep learning based image classification methods.

- The ideas pursued in the work make sense and the particular combination of methods and problem has not been seen before as far as I know.

- The method is compared to a good number of reference methods and seems to improve on them.

Weaknesses
----------
- The two datasets does not seem particularly well chosen for this experiment. The Pneumonia being worse than the Colorectal cancer dataset, but both seem to converge already after 10 added "annotations". As this appears to be the first point on the plot with information, it does not illustrate the convergence properties of the different methods well. As the iterations after the first 10 "annotations" do not seem to improve results, I am wondering what the role of the reinforcement learning aspect is and how much the improvement in performance of the proposed method is simply due to the implemented combination of uncertainty and diversity based sampling.

- I find the novelty to be a bit weak. Casanova et al., 2020 (referenced in paper) already explored very similar techniques for segmentation. An experiment with classification and slightly different approach is a valid contribution if the experimentation was done right, but I am not sure what to conclude from the results I see and the authors comments are not helpful (see comment above).

- Why does the test accuracy seem to have a lower bound at 0.6 for the pneumonia dataset?

- The paragraphs at the bottom of page 3, and equations 1 and 2 are a bit unclear to me. Could you elaborate and provide some intution for the content of these equations?



**Summary Of The Paper:**

The paper proposed a deep learning based reinforcement active learning method for image classification. The approach uses the deep-Q network for learning to pick informative data points. The contributions are a method that allows for the selection of multiple data points in each round, to be more amenable to batch based optimization. Points that maximize uncertainty and diversity is prioritized. The approach is evaluated on two datasets, one containing chest X-rays with the goal of classifying pneumonia and the other being histopathology images of patients with colorectal cancer.


**Summary Of The Review:**

I am really not sure what to make of this work. It could be good with more work, but as it is, there are just too many uncertain aspects.

---

### Decision · Program_Chairs · 2023-01-20

**Decision:**

Reject

**Justification For Why Not Higher Score:**

Because of the weaknesses of the paper, all reviewers agree on rejection.

**Justification For Why Not Lower Score:**

N/A

**Metareview: Summary, Strengths And Weaknesses:**

All reviewers find that the paper lacks novelty.
All reviewers point out significant weaknesses in the experimental evaluation.
No author response has been submitted.